# Association of Reduced Free T3 to Free T4 Ratio with Lower Serum Creatinine in Japanese Hemodialysis Patients

**DOI:** 10.3390/nu13124537

**Published:** 2021-12-17

**Authors:** Masaaki Inaba, Katsuhito Mori, Yoshihiro Tsujimoto, Shinsuke Yamada, Yuko Yamazaki, Masanori Emoto, Tetsuo Shoji

**Affiliations:** 1Renal Center, Ohno Memorial Hospital, 1-26-10, Minami-Horie Nishi-ku, Osaka 550-0015, Japan; 2Department of Nephrology, Osaka City University School of Medicine, 1-4-3, Asahi-machi, Abeno-ku, Osaka 545-8585, Japan; ktmori@med.osaka-cu.ac.jp; 3Division of Internal Medicine, Inoue Hospital, 16-17 enoki-machi, Osaka 564-0053, Japan; tujimoto.yoshihiro@aijinkai-group.com; 4Department of Metabolism, Endocrinology, and Molecular Medicine, Osaka City University School of Medicine, 1-4-3, Asahi-machi, Abeno-ku, Osaka 545-8585, Japan; s.yamada@med.osaka-cu.ac.jp (S.Y.); y.yamazaki911@gmail.com (Y.Y.); memoto@med.osaka-cu.ac.jp (M.E.); 5Vascular Science Center for Translational Research, Osaka City University Graduate School of Medicine 1-4-3, Asahi-machi, Abeno-ku, Osaka 545-8585, Japan; t-shoji@med.osaka-cu.ac.jp; 6Department of Vascular Medicine, Osaka City University Graduate School of Medicine, 1-4-3, Asahi-machi, Abeno-ku, Osaka 550-0015, Japan

**Keywords:** low T3 syndrome, hemodialysis, sarcopenia, malnutrition, creatinine

## Abstract

Purpose: Low T3 syndrome is defined by a fall in free triiodothyronine (FT3) in spite of normal serum thyroid-stimulating hormone (TSH) and often normal free thyroxin (FT4). A low FT3/FT4 ratio, a relevant marker for low T3 syndrome, is known as a risk of mortality in hemodialysis (HD) patients, as well as low muscle mass in the general population. Because of the local activation of T4 to FT3 in muscle tissue, we examined the association of FT3/FT4 ratio with serum creatinine, a marker of muscle mass and strength in HD patients to investigate the significance of muscle tissue in the development of low T3 syndrome in HD patients. Methods: This was a cross-sectional study derived from our prospective cohort study, named DREAM, of Japanese HD patients. After the exclusion of patients with treated and untreated thyroid dysfunction, 332 patients were analyzed in the study. Results: The serum FT4 and TSH of HD patients (*n* = 332) were 0.9 ± 0.1 ng/dL. and 2.0 ± 0.9 μIU/mL, which were within the respective normal range, while serum FT3 was 2.2 ± 0.3 pg/mL. As many as 101 out of 332 (30.4%) HD patients exhibited a serum FT3 less than the normal lower limit of 2.2 pg/mL. The serum FT3/FT4 ratio correlated significantly positively with serum creatinine, and inversely with serum log CRP and total cholesterol, while it exhibited a tendency towards positive correlation with serum albumin. Multiple regression analysis, which included serum creatinine, albumin, and log CRP, simultaneously, in addition to sex, age, diabetic kidney disease or not, log HD duration, body mass index, systolic blood pressure, and Kt/V, as independent variables, revealed an independent and significant positive association of serum creatinine, but not serum albumin or CRP, with the serum FT3/FT4 ratio. Conclusions: The present study demonstrated an independent and positive correlation of serum creatinine with the serum FT3/FT4 ratio in HD patients. The lack of association of the serum FT3/FT4 ratio with serum albumin and CRP suggested the presence of a creatinine-specific mechanism to associate with serum FT3/FT4 ratio. Because of the local activation of T4 to T3 at muscle tissue, a lower muscle mass may be causatively associated with low T3 syndrome.

## 1. Introduction

Chronic kidney disease (CKD) is often associated with an alteration in thyroid hormone status, leading to the development of euthyroid sick syndrome or low triiodothyronine (T3) syndrome [1,2], the latter of which is characterized by a fall in free T3 (FT3) in spite of serum levels of thyroid-stimulating hormone (TSH) and free thyroxin (FT4) maintained within normal ranges [3]. Although TSH has been considered to be the most sensitive single measure of thyroid function for accurate classification of hypothyroidism [4], it is increasingly being recognized that, because of its direct biologic action [5], a reduction in serum FT3 may indicate hypothyroidism in CKD and ESRD patients in spite of normal TSH [6,7]. Although its significance is generally under-recognized, low T3 syndrome is highly prevalent in dialysis patients [8], and it has been speculated that this syndrome is a physiologic adaptation to conserve metabolism in dialysis patients prone to hypercatabolism, dialytic protein and amino acid losses, and protein-energy wasting [9,10], by decreasing T3-stimulated metabolism [5,11]. The mechanism by which serum FT3 decreases relative to serum FT4 in spite of normal TSH is mainly explained by a decreased T4 to T3 deiodination process caused by various factors, such as inflammation and malnutrition [12], mainly in skeletal muscle [13]. Furthermore, it was recently shown in an aged population that sarcopenia might be associated with reduced T4 to T3 deiodination due possibly to smaller site of deiodination [14]. The rate of the process of T4 to T3 deiodination in low T3 syndrome cases is known to be represented by the serum FT3/FT4 ratio [14,15],

The present cross-sectional study examined (i) the prevalence of low T3 syndrome in Japanese hemodialysis patients, and (ii) the identification of the most important factor among sarcopenia, malnutrition, or inflammation to associate with a reduced serum FT3/FT4 ratio in TSH-normal hemodialysis patients.

## 2. Materials and Methods

### 2.1. Selection of Subjects

The present study was conducted as a cross-sectional analysis of our prospective study cohort termed DREAM (Dialysis-Related Endocrine And Metabolic changes affecting cardiovascular disease) [11,16,17]. The total of 806 dialysis patients were treated at Inoue Hospital, Suita, Osaka, Japan, in December 2004. The patients who were maintained on peritoneal dialysis (*n* = 71), hospitalized (*n* = 31), suffering from dementia-disability (*n* = 99), in addition to those who declined to participate (*n* = 87), were excluded from the study. The DREAM cohort included 518 patients stably maintained on hemodialysis (HD) at Inoue Hospital who were examined in December 2004. Those patients were undergoing four-hour HD sessions three times per week at the outpatient clinic of Inoue Hospital Kidney Center, Japan. The protocol of the present study was approved by the Ethics Review Committee of Inoue Hospital (approval #121) and it was conducted in accordance with the principles of the Declaration of Helsinki. The DREAM cohort has been registered at UMIN-CTR (ID UMIN000006168, http://www.umin.ac.jp/ctr/index.htm accessed on 17 December 2021). Written informed consent was obtained from each participant prior to participation.

To select HD patients eligible for the present study, only those with normal serum TSH (0.4–4.0 μIU/mL) were selected for the present study. We excluded the following patients: (1) those treated for hyperthyroidism (*n* = 2) and hypothyroidism (*n* = 37), (2) those with corticoid use (*n* = 21), (3) those with primary hyperthyroidism (*n* = 5) and hypothyroidism (*n* = 120), (4) those with secondary hyperthyroidism (*n* = 0) and hypothyroidism (*n* = 0), and those with missing data on serum FT3, FT4, or TSH (*n* = 1) (Figure 1).

### 2.2. Measurement of Thyroid Hormones in Serum

Blood samples were drawn from the arteriovenous fistula just prior to an HD session at the beginning of the week, three days after the previous HD session, as previously reported [18]. Serum levels of FT3, FT4, and TSH were measured by specific chemiluminescent immunoassay (CLIA) methods performed by BIO MEDICAL LABORATORIES (BML, Inc., Tokyo, Japan), as previously described [11]. The reference ranges for FT3, FT4, and TSH were 2.2–4.1 pg/mL, 0.8–1.9 ng/dL, and 0.4–4.0 μIU/mL, respectively.

## 3. Statistical Analysis

Continuous variables with normal and non-normal distribution are expressed as the mean ± SD, and median and interquartile range (IQR), respectively. Correlation coefficients between the serum FT3/FT4 ratio and various clinical parameters were calculated by simple regression analysis. Multivariable regression analyses were performed to evaluate the associations of the serum FT3/FT4 ratio with various clinical parameters. Serum CRP levels and HD duration were logarithmically transformed due to the skewed distribution. Statistical analysis was performed using the StatView V system for Windows (Abacus Concepts, Inc., Berkeley, CA, USA) on a personal computer. *p* values < 0.05 were considered to indicate statistical significance.

## 4. Results

### 4.1. Selection of HD Patients for Analysis

As shown in Figure 1, among all 518 maintenance HD patients in the DREAM cohort, 332 patients, who were assessed to be in a euthyroid state based on a serum TSH level between 0.4–4.0 μIU/mL, were included in the present study. The serum FT3 in 231 of those patients was within a normal range (2.2–4.1 pg/mL), while the remaining 101 patients exhibited a serum FT3 level lower than the lower normal limit of 2.2 pg/mL, indicating that 101 out of 332 HD patients may have suffered from low T3 syndrome.

### 4.2. Clinical Characteristics of Euthyroid HD Patients

The clinical characteristics of the 332 euthyroid HD patients (mean age 59.9 ± 11.5 years, M/F:214/118) analyzed for the present study are shown in Table 1. The prevalence of DM was 23.5%. Serum TSH, FT4, and FT3 levels were 2.0 ± 0.9 μIU/mL, 0.9 ± 0.1 ng/dL, and 2.2 ± 0.3 pg/mL, respectively, while the serum FT3/FT4 ratio [(pg/mL)/(ng/dL)] was 2.3 ± 0.5.

### 4.3. Correlation of Serum FT3/FT4 Ratio with Various Clinical Parameters

The serum FT3/FT4 ratio was examined for its association with various clinical parameters in the 332 euthyroid HD patients, and found to be significantly correlated in a positive manner with serum creatinine (Figure 2A) and in a negative manner with serum log CRP (Figure 2B). Furthermore, that ratio showed a tendency to have a positive correlation with serum albumin (Figure 2C). It is of interest that the serum FT3/FT4 ratio showed a significant and inverse correlation with serum total cholesterol (Figure 2D), as well as LDL-cholesterol, though serum total cholesterol, as well as serum albumin and creatinine, is known to be a definite nutritional marker in HD patients.

### 4.4. Multiple Regression Analysis of Serum Creatinine, Albumin, and CRP with Serum FT3/FT4 Ratio

Multiple regression analysis was performed to examine whether serum creatinine, serum albumin, or log CRP had a significant association with the serum FT3/FT4 ratio in the present euthyroid HD patients (Table 2). When serum creatinine was included as an independent variable, in addition to gender, age, presence of diabetic kidney disease, log HD duration, BMI, SBP, and Kt/V (Model 1), it emerged as a significant and independent factor with a positive association with the serum FT3/FT4 ratio. When serum creatinine was replaced with serum albumin (Model 2), serum albumin failed to show a significant association with the serum FT3/FT4 ratio, whereas when serum albumin was replaced with serum log CRP (Model 3), serum log CRP showed a significant and inverse association with the serum FT3/FT4 ratio. Serum creatinine retained its independent and significant and positive association with the serum FT3/FT4 ratio even when serum creatinine was simultaneously included with serum albumin and/or log CRP as independent variables in Models 4–7.

## 5. Discussion

The present study demonstrated an independent association of lower serum creatinine, but not of albumin or log CRP, with a lower serum FT3/FT4 ratio in Japanese HD patients (Table 2). Both serum creatinine and albumin are known to be relevant clinical markers of nutritional state in HD patients [19], which is supported by the present findings of a significant and positive correlation between serum creatinine and albumin (r = 0.388, *p* < 0.0001). Furthermore, results showing that creatinine was significantly and inversely correlated with serum log CRP in the present cohort (r = −0.194, *p* = 0.0004) indicated that the serum FT3/FT4 ratio is significantly influenced by both nutrition and inflammation in HD patients, as previously described [20]. In multivariate models that included serum creatinine, albumin, and log CRP as independent variables, serum creatinine alone retained a significant and positive association with the serum FT3/FT4 ratio in each of those models (Models 4–7, Table 2), suggesting that the independent association of serum creatinine with the serum FT3/FT4 ratio is by a mechanism other than nutritional and inflammation factors.

In unhealthy patients, such as those with CKD, thyroid function is known to fall in order to reduce energy expenditure [9,10] caused by a fall in serum FT3 in cases with normal TSH and mostly normal FT4, termed low T3 syndrome, or those with nonthyroidal illness syndrome [2,3,21]. Although TSH is considered to be the most sensitive marker for thyroid function [4], low circulating FT3 is the most frequently encountered thyroid functional test abnormality seen in CKD patients [11]. The HD patients enrolled in the present study were restricted to those with normal TSH, though it is becoming increasingly recognized that serum FT3 is more important than serum TSH for evaluating thyroid gland function because of its direct action in the target tissue [22]. A previous study also showed that the serum FT3/FT4 ratio clearly indicates a reduction in thyroid function in affected patients and the severity of low T3 syndrome [23]. In support of this notion, the present findings demonstrated that a reduction in FT3/FT4 ratio clearly revealed reduced thyroid function as shown by the significant and inverse correlation with serum total cholesterol (Figure 2D), but not with serum creatinine or albumin (Figure 2A,C). Only serum total cholesterol, not albumin or creatinine, is regulated negatively by thyroid function [24], thus the inverse correlation of the serum FT3/FT4 ratio with serum cholesterol, but not with serum albumin or creatinine, noted here clearly indicates that the serum FT3/FT4 ratio is a reliable marker for thyroid function. However, low serum cholesterol is often observed in malnourished HD patients, the inverse correlation between two parameters cannot be solely by low T3-induced hypothyroidism.

Serum creatinine is known to increase under hypothyroidism mainly by a reduction in renal function [25], while we previously reported that serum creatinine was decreased with hyperthyroidism in HD patients without apparent residual renal function [26]. Therefore, neither reduced renal function nor thyroid hormone-induced suppression of serum creatinine cannot explain the association of lower serum creatinine with a lower FT3/FT4 ratio in HD patients. In the present study, serum creatinine was lower in association with a reduced serum FT3/FT4 ratio, indicating that it should be determined mainly by factors other than thyroid hormone status.

We and others have demonstrated that serum creatinine is a clinically relevant marker for muscle mass and strength in HD patients [27,28]; thus, it is reasonable to consider that reduced muscle mass and/or strength are factors intimately associated with a reduced FT3/FT4 ratio in HD patients. Supportive of this notion are previous reports showing that a reduced FT3/FT4 ratio was positively associated with low muscle mass and impaired physical performance in aged euthyroid populations [14,15]. Skeletal muscle is known as a major target tissue of thyroid hormones [29]. It has also been reported that type 2 iodothyronine deiodinase (DIO2) in skeletal muscle mainly converts T_4_ to biologically active T_3_ by outer ring monodeiodination of T4, which contributes mostly to the intracellular T_3_ pool in skeletal muscle [30]. T3 initially generated inside muscle cells may exit to mix with the extracellular pool, thus increasing FT3 in serum, which suggests that greater amounts of T3 generated by DIO2 inside skeletal myocytes might increase serum FT3 by moving to outside of the cells. In a previous study, a 70-kg euthyroid subject was found to produce a total extrathyroidal T3 production of 44 nmol/day [31], among which the amount of hepatic DIO1-catalyzed T_4_ to T_3_ conversion was approximately 15 nmol of T_3_/day while that of DIO2-catalyzed T_3_ production from T_4_ in skeletal muscle was approximately 29 nmol/day [32]. Therefore, because of the two-to-threefold greater local production of T_3_ in skeletal muscle than in the liver, it was suggested that the impairment of DIO2-generated T3 in skeletal muscle resulting from the decrease in muscle mass may be a major cause of the reduced serum FT3 in euthyroid sick syndrome. These results indicate that the mechanism by which serum creatinine, but not serum CRP or albumin, is significantly associated with serum FT3/FT4 ratio, could be explained by the notion that sarcopenic HD patients exhibit lower serum FT3 by impaired local T3 generation from T4 in muscle mass resulting from lower levels of muscle mass and/or muscle activity. Furthermore, it is possible that hypometabolism in sarcopenia patients might be explained, in part, by impaired T4 deiodination in muscle tissue [33], and that maintenance of muscle mass in HD patients may protect against the development of low T3 syndrome, and thus improve quality of life and mortality.

This study has some limitations. First, the sample size was small and all subjects had Japanese ethnicity. Second, we measured thyroid function only once. Third, we determined thyroid function only by serum levels of thyroid hormones and did not obtain collective information related to thyroid disease, such as goiter and thyroid-related auto-antibodies. Fifth, the results of a cross-sectional study design cannot indicate causality. Sixth, serum FT3/FT4 ratio could be determined not only by serum T3 but also by serum T4, it is yet to be determined that the reduction of serum FT3/FT4 ratio could definitely indicate low-grade-hypothyroidism although we recruited the HD patients only with normal serum FT4. On the other hand, the strong points of this study included its performance by a single institution and the same dialysis condition with the same dialysate solution under a controlled situation managed by the same staff.

In conclusion, the present results clearly indicate (i) a high prevalence of low T3 syndrome in Japanese HD patients and (ii) an association of a lower serum FT3/FT4 ratio with lower serum creatinine. The association between those two parameters was found to be independent of nutrition and inflammation; thus, it was concluded that sarcopenia in HD patients might cause hypometabolism by inducing low T3 syndrome by decreasing T4 to T3 activation in muscle tissue. In consideration of the suppression of T3 as a risk factor for cardiovascular events and mortality [1,34], prevention of the development of sarcopenia in HD patients is suggested to be an important issue in HD patients from this standpoint.

## Figures and Tables

**Figure 1 nutrients-13-04537-f001:**
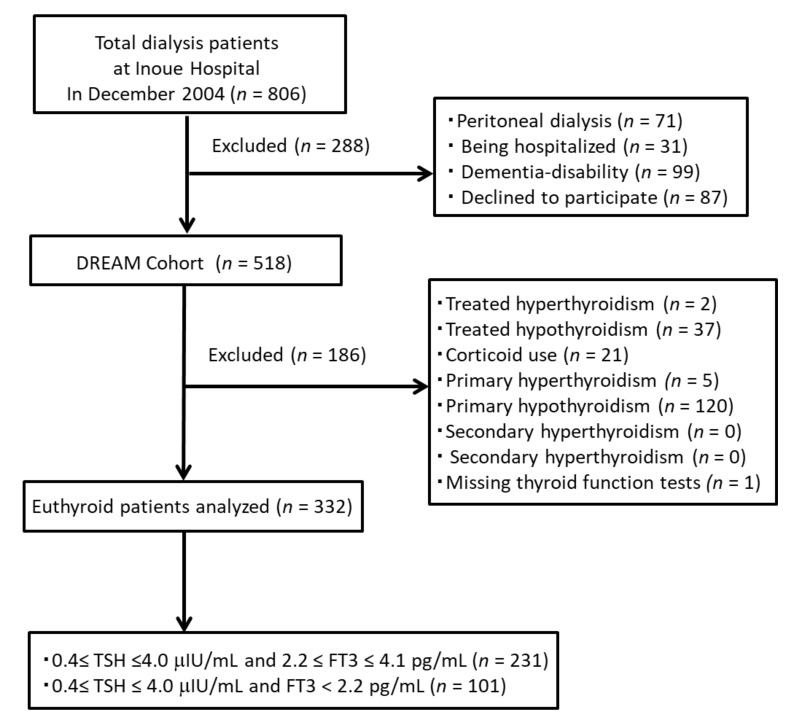
Selection of subjects for present study.

**Figure 2 nutrients-13-04537-f002:**
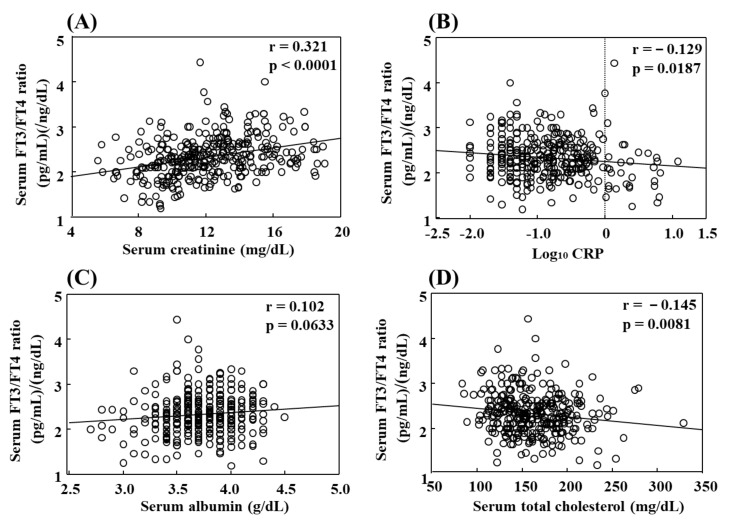
Correlation of serum FT3/FT4 ratio with (**A**) serum creatinine, (**B**) log10 CRP, (**C**) serum albumin, and (**D**) serum total cholesterol in 332 hemodialysis patients. Serum FT3/FT4 ratio was significantly correlated in a positive manner with serum creatinine (**A**) and in a negative manner with serum log10 CRP (**B**). In addition, it showed a tendency towards a positive correlation with serum albumin (**C**), while a significant and inverse correlation with serum total cholesterol was found (**D**).

**Table 1 nutrients-13-04537-t001:** Baseline clinical characteristics of 332 euthyroid hemodialysis patients.

*n*	332
Gender (M/F)	214/118
Age (years)	59.9 ± 11.5
DM, n (%)	78 (23.5)
HD duration, [months (IQR)]	101.5 (42.0–171.5)
BMI (kg/m^2^)	22.1 ± 2.9
Systolic BP (mmHg)	151.9 ± 18.0
Diastolic BP (mmHg)	78.0 ± 7.4
Kt/V	1.4 ± 0.3
Cardiothoracic ratio (%)	49.0 ± 4.9
Creatinine (mg/dL)	12.2 ± 2.8
Total Protein (g/dL)	6.6 ± 0.5
Albumin (g/dL)	3.7 ± 0.3
CRP [mg/dL (IQR)]	0.14 (0.05–0.34)
Total cholesterol (mg/dL)	161.8 ± 35.3
HDL cholesterol (mg/dL)	46.0 ± 13.9
LDL cholesterol (mg/dL)	91.0 ± 28.1
PTH intact [pg/mL(IQR)]	120.0 (42.5–219.0)
TSH (μIU/mL)	2.0 ± 0.9
FT4 (ng/dL)	0.9 ± 0.1
FT3 (pg/mL)	2.2 ± 0.3
FT3/FT4 ratio [(pg/mL)/(ng/dL)]	2.3 ± 0.5

**Table 2 nutrients-13-04537-t002:** Multivariate analysis of factors associated with serum FT3/FT4 ratio in euthyroid hemodialysis patients (*n* = 332).

	FT3/FT4 Ratio						
	Model 1	Model 2	Model 3	Model 4	Model 5	Model 6	Model 7
Gender (m/f = 1/2)	−0.230 *	−0.285 *	−0.292 *	−0.231 *	−0.241 *	−0.292 *	−0.240 *
Age (years)	−0.108	−0.179 *	−0.163 *	−0.106	−0.091	−0.163 *	−0.097
DM (non-DM/DM = 0/1)	−0.022	−0.058	−0.056	−0.021	−0.022	−0.056	−0.023
Log HD duration	−0.017	−0.003	0.006	−0.016	−0.006	0.006	−0.008
BMI	0.049	0.074	0.093	0.050	0.071	0.093	0.069
SBP	−0.117 *	−0.106	−0.100	−0.118 *	−0.116 *	−0.099	−0.111
Kt/V	0.072	0.066	0.061	0.072	0.069	0.061	0.068
Creatinine	0.171 *	―	―	0.169 *	0.153 *	―	0.157 *
Albumin	―	0.033	―	0.005	―	−0.001	−0.023
Log CRP	―	―	−0.109 *	―	−0.092	−0.109	−0.098
r^2^ (*p*)	0.161 (<0.0001)	0.147 (<0.0001)	0.157 (<0.0001)	0.161 (<0.0001)	0.168 (<0.0001)	0.157 (<0.0001)	0.169 (<0.0001)

* Statistically significant (*p* < 0.05).

## Data Availability

The data that support the findings of this study are available on request from the corresponding author. The data are not publicly available due to privacy or ethical restrictions.

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
