# Peer review of "Association of Reduced Free T3 to Free T4 Ratio with Lower Serum Creatinine in Japanese Hemodialysis Patients"

_nutrients, 2021, doi:10.3390/nu13124537_

Round 1

Reviewer 1 Report

This is an interesting paper but I have some concerns.

  1. Though the conclusions are reasonable and plausible they remain hypothetical. The title of the paper indicates an association with sarcopenia. There is no direct documentation of sarcopenia to provide empiric evidence- all we have is an association with creatinine which in turn is associated with muscle mass. There may be other associations that are more relevant or the actual cause of the association ( eg were  the effects of exercise on thyroid function and muscle mass a possibility i.e. did exercise increase thyroid function independently of its effect on muscle mass). The authors should limit their conclusions to their evidence or at least in the title (if they wish to keep that title) indicate that the conclusions are hypothetical.
  2. The studied population is from 2004. the authors should explain the reason for using such old data.
  3.  There should be more  data, analysis and discussion of FT4.  There should in fact be analyses of F4 and FT3 so that FT3/FT4 ratio can be shown to be superior to either. Recent work shows that clinical parameters are more associated with FT4 than with TSH. There were similar degrees of associations with FT3 but these were often in the context of suspected reverse causation. In this regard I note that the baseline FT4 in the group was 0.9 - i.e. at the lower edge of the normal range.
  4. If these concerns can be addressed or rebutted the paper may be stronger.

Author Response

This is an interesting paper but I have some concerns.

1. Though the conclusions are reasonable and plausible they remain hypothetical. The title of the paper indicates an association with sarcopenia. There is no direct documentation of sarcopenia to provide empiric evidence- all we have is an association with creatinine which in turn is associated with muscle mass. There may be other associations that are more relevant or the actual cause of the association ( eg were  the effects of exercise on thyroid function and muscle mass a possibility i.e. did exercise increase thyroid function independently of its effect on muscle mass). The authors should limit their conclusions to their evidence or at least in the title (if they wish to keep that title) indicate that the conclusions are hypothetical.

Thank you for your valuable comment.  Since we demonstrated an independent association of serum FT3/FT4 ratio with serum creatinine, but not with muscle mass/strength, we agree to change the title from “Association of reduced free T3 to free T4 ratio with sarcopenia in Japanese hemodialysis patients” to “Association of reduced free T3 to free T4 ratio with reduced serum creatinine in Japanese hemodialysis patients”

2. The studied population is from 2004. the authors should explain the reason for using such old data.

We recently found that low T3 syndrome may be explained by decreased muscle mass because of reduced activation of thyroid hormone activation from T4 to T3 at muscle.  Although we have continued to renew clinical database of hemodialyzed patients, the database which containing thyroid hormone data was that made in 2004.  However, the size of this database is sufficient to analyze the association of low T3 syndrome with serum creatinine and various clinical parameters in hemodialysis patients as shown in Table 1.

3. There should be more data, analysis and discussion of FT4.  There should in fact be analyses of FT4 and FT3 so that FT3/FT4 ratio can be shown to be superior to either. Recent work shows that clinical parameters are more associated with FT4 than with TSH. There were similar degrees of associations with FT3 but these were often in the context of suspected reverse causation. In this regard I note that the baseline FT4 in the group was 0.9 - i.e. at the lower edge of the normal range.

Thank you for your valuable suggestion. I agreed with your opinion that the decreased serum FT4 and FT3 by itself is important to analyze its association with serum creatinine and various clinical parameters in hemodialysis patients. However, the aim of the present study focused on the reduced thyroid hormone activation from T4 to T3 at the muscle resulting form the reduced muscle mass as reflected by reduced serum creatinine in hemodialysis patients without apparent residual renal function. Therefore, we restricted of enrolled patients with their serum FT4 within normal range.  In such cases, it is reported that serum FT3/FT4 ratio could provide the clinically relevant measure of low T3 syndrome.  So, we believe that the analysis of serum FT3/FT4 ratio with serum creatinine and various clinical parameters might provide evidence for the association of reduced activation of T4 to T3 with reduced muscle mass as reflected by reduced serum creatinine in hemodialysis patients.

 I will continue my work as you suggested as another project.

4. If these concerns can be addressed or rebutted the paper may be stronger.

Thank you very much again for your valuable comments.

Reviewer 2 Report

This is a cross-sectional on TSH-normal hemodialysis patients aiming to assess the prevalence of low T3 syndrome in these patients and the factors predisposing to it. The prevalence is reported to be increased and a strong correlation of lower serum FT3/FT4 ratio with lower serum  creatinine, independent of malnutrition and inflammation markers is found.

The design and realization of the study are sound (there are some minor limitations acknowledged by the authors), the area of research is very interesting and only scarcely investigated so far.  The study should be continued and developed in the future including more patients and specific investigations for sarcopenia (proposed as the link mechanism to explain the findings of the study) but, given the scarcity of literature in this area and the interesting conclusions, in my opinion at present the study can be considered for publication.

Author Response

This is a cross-sectional on TSH-normal hemodialysis patients aiming to assess the prevalence of low T3 syndrome in these patients and the factors predisposing to it. The prevalence is reported to be increased and a strong correlation of lower serum FT3/FT4 ratio with lower serum creatinine, independent of malnutrition and inflammation markers is found.

The design and realization of the study are sound (there are some minor limitations acknowledged by the authors), the area of research is very interesting and only scarcely investigated so far.  The study should be continued and developed in the future including more patients and specific investigations for sarcopenia (proposed as the link mechanism to explain the findings of the study) but, given the scarcity of literature in this area and the interesting conclusions, in my opinion at present the study can be considered for publication.

Thank you for your wonderful comments.  We are encouraged to continue this kind of work in future.  Thank very much again.
